# Cyclodextrin Multicomponent Complexes: Pharmaceutical Applications

**DOI:** 10.3390/pharmaceutics13071099

**Published:** 2021-07-20

**Authors:** Virginia Aiassa, Claudia Garnero, Marcela R. Longhi, Ariana Zoppi

**Affiliations:** Unidad de Investigación y Desarrollo en Tecnología Farmacéutica, UNITEFA-CONICET, Departamento de Ciencias Farmacéuticas, Facultad Ciencias Químicas, Universidad Nacional de Córdoba, Córdoba 5000, Argentina; viraiassa@unc.edu.ar (V.A.); cgarnero@unc.edu.ar (C.G.); azoppi@unc.edu.ar (A.Z.)

**Keywords:** auxiliary agents, complexation efficiency amino acids, organic acids, organic bases, water-soluble polymers

## Abstract

Cyclodextrins (CDs) are naturally available water-soluble cyclic oligosaccharides widely used as carriers in the pharmaceutical industry for their ability to modulate several properties of drugs through the formation of drug–CD complexes. The addition of an auxiliary substance when forming multicomponent complexes is an adequate strategy to enhance complexation efficiency and to facilitate the therapeutic applicability of different drugs. This review discusses multicomponent complexation using amino acids; organic acids and bases; and water-soluble polymers as auxiliary excipients. Special attention is given to improved properties by including information on the solubility, dissolution, permeation, stability and bioavailability of several relevant drugs. In addition, the use of multicomponent CD complexes to enhance therapeutic drug effects is summarized.

## 1. Introduction

Cyclodextrins (CDs) are a family of macromolecules composed of α-1,4-linked D-glucopyranoside subunits obtained by enzymatic degradation of starch [1]. Their structure resembles a truncated cone, with a somewhat lipophilic central cavity and an external hydrophilic surface. Antoine Villiers discovered the crystalline dextrins that were later recognized as CDs, and he published the first reference to these compounds in 1891 [2]. The most common natural CDs are αCD, βCD and γCD (Figure 1), consisting of 6, 7 and 8 D-glucose units, and they are the only ones used for pharmaceutical applications. Several researchers have attempted to develop other CD derivatives of pharmaceutical interest. Chemically modified CDs can be obtained from the introduction of different functional groups at the 2, 3 and 6-hydroxyl groups of the glucose residues. These include, for example, the hydroxypropyl, methylated and sulfobutylether β-cyclodextrins (HPβCD, MβCD, SBEβCD), and (2-hydroxy)propyl-γ-cyclodextrin (HPγCD), among others [3].

CDs have been used in pharmaceutical formulations since the 1970s. The first pharmaceutical product containing CD, marketed in Japan in 1976, was Prostarmon E™ (prostaglandin E2:βCD) [4]. Currently, there are various commercial formulations containing CDs. In addition, CD monographs are included in various pharmacopoeias and are accepted as pharmaceutical excipients by various regulatory agencies. The special interest for CDs in the field of pharmaceutical applications is due to their ability to modulate several properties affecting the performances and therapeutic profiles of drugs. CDs can be used to enhance the aqueous solubility of poorly soluble drugs; increase drug permeability through biological membranes; improve drug bioavailability and stability; reduce adverse drug reactions, such as gastrointestinal or ocular irritation and other side effects; convert liquids and oils into free-flowing powders; reduce evaporation and stabilize flavors; improve palatability and handling; and prevent admixture incompatibilities [5]. It should be noted that the modified γ-CD sugammadex, developed by the pharmaceutical company Organon, is used as a drug to reverse the neuromuscular blockade induced by rocuronium and vecuronium in adults undergoing surgery [6].

All these pharmaceutical applications are mostly related to the ability of CDs to form inclusion complexes. An inclusion complex is formed by interactions between guest (drug) and host (CD) molecules, and thus requires structural complementarity between the interacting compounds. A drug–CD complex that can be optimized with the addition of an auxiliary substance can be called a multicomponent or ternary complex. In these cases, a synergic effect is desired to allow the use of low concentrations of the non-drug compounds. The use of multicomponent complex formation with CDs and a third auxiliary substance has become more frequent in order to facilitate the therapeutic applicability of different drugs. The incorporation of small amounts of various additives, such as amino acids; organic acids and bases; and water-soluble polymers in aqueous complexation media can increase the complexation efficiency, which permits the use of considerably lower quantities of CDs, thereby optimizing the cost, toxicity and formulation volume in the final product [7,8,9,10]. These additives might also interact with CDs in order to improve and modulate the drugs’ physicochemical and transport properties. This interaction can modulate in vitro and in vivo drug dissolution, thereby modifying the drug’s pharmacokinetic profile. Moreover, it is possible to reduce other problematic behaviors of the drugs, such as their toxicological properties, by specifically selecting an auxiliary substance that will be added to the multicomponent complex. Regarding the methodological aspects, the techniques commonly used to obtain binary complexes can be used in the preparation of ternary ones in solid state. Methods such as blending, co-grinding, kneading, coevaporation/coprecipitation, freeze-drying, spray-drying and supercritical fluid technology have been used [8]. The method used for the multicomponent complex’s preparation has significant impacts on the final product. Depending on the method selected, different parameters should be optimized, such as the intensity of mixing, mixing time, temperature and heating time.

Since amino acids; organic acids and bases; and water-soluble polymers have some advantages over other excipients when used as auxiliary agents, this article reviews and tries to cover the most relevant multicomponent CD complexes obtained with these auxiliary compounds, along with their uses in the pharmaceutical field. Their applications in drug delivery are illustrated and discussed in depth by using specific examples reported in the literature. These examples support their use in the development of drug formulations with improved properties.

## 2. Amino Acids as Auxiliary Agents

Amino acids (AAs) present a series of adequate properties for pharmaceutical applications, such as a large variety of molecular structures and physicochemical properties, a wide scope of functional groups, low molecular weight and good solubility in aqueous media. In addition, AAs are generally regarded as safe and have great potential for interacting with CDs and drugs via hydrogen bonding with CDs and via electrostatic interactions and salt formation with drugs. Therefore, there is growing interest in utilizing AAs for multicomponent CD complex development to improve unfavorable drug properties. Several examples of multicomponent complexes of CD can be found in the literature in which AAs have been used as auxiliary agents not only to improve the solubility and dissolution rates of poorly soluble drugs, but also to allow the modulation of other drug properties. For example, AAs have shown their usefulness by increasing antimicrobial activity and reducing the toxicity of drugs [7,11].

Among the AAs that have been used to prepare multicomponent complexes, it is possible to mention glycine, cysteine, proline, arginine, lysine, aspartic acid and glutamic acid (Figure 2)—arginine being one of the AAs that has produced the best results in the formation of multicomponent complexes (Table 1, Figure 3). For example, Mura et al. studied the ability of several amino acids (arginine, isoleucine, lysine and valine) in combination with HPβCD to improve the solubility of naproxen, a poorly water-soluble drug. Arginine was the amino acid that allowed the greatest increase in drug solubility, showing a synergistic effect with HPβCD [12]. In addition, the use of arginine in combination with HPβCD improved naproxen’s dissolution rate [13]. Similar results with respect to increased solubility and dissolution were obtained with other nonsteroidal anti-inflammatory drugs, such as lornoxicam, oxaprozin and zaltoprofen, by using a ternary complex with arginine and a CD (βCD, HPβCD or MβCD) [14,15,16]. Besides, arginine was the best tertiary component in regard to improving the solubility and dissolution performance of etodolac, another nonsteroidal anti-inflammatory drug, in the presence of HPβCD [17]. Using computational modeling, the authors demonstrated that arginine forms a bridge between the drug and the HPβCD, through hydrogen bonding and electrostatic interactions, which led to an increase in the stability of the complex. By means of molecular modeling studies, other reports also concluded that arginine plays an important role in the formation of drug–CD multicomponent complexes, and demonstrated the ability of arginine to establish hydrogen bonds with the macromolecule [18,19].

Singh et al. studied combinations of different auxiliary agents (lactose, polyvinylpyrrolidone K-30, poloxamer-188, sodium lauryl sulphate and arginine) with HPβCD to select the most appropriate substance with which to prepare a ternary complex of glyburide. Glyburide is a poorly water-soluble drug used as a hypoglycemic agent. The drug’s solubility was studied in an aqueous solution and in a pH 7.5 phosphate buffer system; in both media, the ternary complex glyburide–HPβCD–arginine presented superior solubility compared with the other binary and ternary systems. The stability constant values reported at pH = 7.5 were 100 M^−1^ for the binary complex and 360 M^−1^ for the ternary one [20]. Repaglinide and nateglinide belong to the meglitinide class of blood glucose-lowering drugs. In order to improve the solubilities and dissolution rates of these Biopharmaceutical Classification System (BCS) class II drugs, the effects of different auxiliary agents were studied in combination with HPβCD. Arginine was the best candidate for forming ternary complexes with both drugs. The authors found the following values for stability constant and complexation efficiency: repaglinide–HPβCD, 333 M^−1^ and 0.025; repaglinide–HPβCD–arginine, 4407 M^−1^ and 0.340; nateglinide–HPβCD, 382 M^−1^ and 0.435; nateglinide–HPβCD–arginine, 464 M^−1^ and 0.526 [21,22].

Cefixime is a third-generation cephalosporin that has poor aqueous solubility and variable bioavailability. Jadhav et al. proved that the formation of cefixime–CD–arginine ternary complexes constitutes a good strategy to enhance the solubility and dissolution of the drug. The use of arginine allows increasing the slope, stability constant and complexation efficiency with both CDs (βCD and HPβCD). The binary and ternary systems prepared by the spray drying method showed better dissolution rates than the drug alone; the ternary complex with HPβCD showed the best results [23]. Rifampicin is another antibiotic drug with low water solubility and variable bioavailability. Dan Córdoba et al. demonstrated that multicomponent complex formation with βCD and arginine was an adequate approach to improving not only the solubility and the dissolution profile of rifampicin, but also its antibiofilm activity against methicillin-sensitive and methicillin resistant *Staphylococcus aureus* bacterial strains. Using theoretical studies, the authors found that arginine can establish hydrogen bond interactions with the drug and βCD, which leads to an increase in the stability of the multicomponent complex [24].

Conde Penedo et al. studied the effect of a combination containing HPβCD, chitosan and arginine on the solubility and corneal permeability of riboflavin. The pure drug has a solubility of 0.12 mg/mL, whereas in the presence of HPβCD or HPβCD–chitosan–arginine, the solubility was 0.35 or 0.78 mg/mL, respectively. In addition, this combination had a synergistic effect on the permeability of the drug with the absence of toxicity [25].

Mora et al. prepared a 1–2–3 dehydroepiandrosterone–αCD–glycine ternary complex to improve the solubility and bioavailability of this drug. This complex showed better dissolution properties and improved pharmacokinetic parameters in human volunteers, as compared with the unmodified drug. The area under curve (59 ± 20 and 29 ± 17 ng/mL.h) and the peak serum concentration (C_max_ 10.9 ± 6.3 and 4.8 ± 3.8 ng/mL) for the complex were higher than those obtained with the drug alone, and the time to reach C_max_ was reduced (T_max_ 0.5 ± 0.2 and 2.2 ± 0.9 h for the complex versus drug alone, respectively) [26].

Piel et al. evaluated the effect of lysine in combination with βCD and γCD on the aqueous solubility of nimesulide, a nonsteroidal anti-inflammatory drug with very poor water solubility. The solubility of nimesulide in the presence of lysine–βCD was 6.59 mg/mL, and in the presence of lysine–γCD it was 19.07 mg/mL. These values are greater than the ones obtained using binary systems (nimesulide–lysine 4.45 mg/mL, nimesulide–βCD 0.025 mg/mL and nimesulide–γCD 0.022 mg/mL), evidencing that CDs and lysine have a synergistic effect on nimesulide’s aqueous solubility. Additionally, the solubility of the complexes prepared by precipitation was measured in water and at different pH levels. The nimesulide–βCD–lysine complex was the most soluble [27]. For example, at pH 6.8, the solubility was 2.37 mg/mL for the nimesulide–βCD–lysine complex, compared with 0.015 mg/mL for pure nimesulide and 1.71 mg/mL for the complex nimesulide–lysine–γCD [27]. In another work, Piette et al. used a multicomponent system with lysine and HPβCD to prepare oral and intravenous solutions of Ro 28-2653, a synthetic inhibitor of matrix metalloproteinases in development, which exhibits poor water solubility. These systems increased the drug solubility of Ro 28-2653. In addition, the absolute bioavailability (compared with the intravenous solution) was about 10 times higher for the drug that was orally administered as a multicomponent complex in solution than for a suspension of the drug alone. The solution showed a higher C_max_ (51.8 vs. 4.8 µg/mL) and a lower T_max_ (3.6 vs. 12.3 h) versus the drug alone, respectively [28]. Furthermore, some investigations showed that the incorporation of lysine increases the inclusion efficiency of drugs, such as piroxicam and ketoprofen, into CDs when supercritical fluid technology is used for the complex preparations [29,30].

Cerutti et al. determined that the formation of a multicomponent complex between nifedipine, a calcium-channel blocking agent, βCD and aspartic acid was an adequate strategy for improving the solubility and dissolution profile of the drug. Pure nifedipine showed a maximum of 30 ± 2% drug dissolution at 180 min, compared with the values for the binary and multicomponent complexes, which were 66 ± 6% and 95.0 ± 0.5%, respectively [31]. Norfloxacin, a fluoroquinolone antibiotic, is a weak base drug that has low solubility in water. To overcome this problem, Ponce Ponte et al. studied the possibility of preparing multicomponent complexes using AAs (glutamic acid, lysine and proline) and HPβCD. In this case, glutamic acid was the best candidate with which to form ternary complexes [32]. Chloramphenicol is an antibacterial drug that is slightly soluble in water and has serious side effects. Aiassa et al. investigated different multicomponent complexes of this drug with CDs (βCD and γCD), AAs (glycine and cysteine) and *N*-acetylcysteine, the *N*-acetyl derivative of the amino acid cysteine. All the multicomponent complexes studied increased the solubility of the drug and proved to be less toxic against leukocytes than the pure drug. In addition, the multicomponent complexes with both CDs and *N*-acetylcysteine were more active against *S. aureus* biofilms than the drug alone [11,33,34]. Ketoconazole is an imidazole antifungal drug with low solubility. Zoppi et al. investigated the possibility that proline and *N*-acetylcysteine act as ternary complexation agents for the binding of ketoconazole with βCD. The increases in drug solubility found for these complexes were of 54, 65 and 459-fold for the binary and ternary complexes containing proline or *N*-acetylcysteine, respectively. Both multicomponent complexes showed antifungal activity significantly greater than the drug alone or the binary complex [35,36]. In addition, the complex containing *N*-acetylcysteine was able to decrease the fungal viability in the biofilms of *C. albicans*.

From the analysis of these works, it is evident that proper selection of the auxiliary agent allows one to not only increase the solubility but also to modify other properties of the drug.

## 3. Organic Acids as Auxiliary Agents

Multicomponent complexes with CDs and different acids have been studied to improve the properties by orders of magnitude of basic drugs in relation to classic drug–CD binary complexes. The ternary complexes are obtained by adding certain low-molecular-weight acids, such as hydroxy carboxylic acids (Figure 4) [9,37,38], to the binary complexes composed of the respective drugs and CDs. These auxiliary acid substances, together with the CDs, improve the physicochemical, chemical and transport properties, and therefore provide advantages in terms of bioavailability and pharmaceutical use [38].

The mechanism of action of weak acids allows stabilizing a ternary complex through a combination of interactions (hydrogen bonding, salt formation and electrostatic interactions) depending on the nature and structural characteristics of the molecules. It is important to note that these acids are compounds rich in hydrophilic groups distributed along their molecular structures that have little affinity for the cavity of the CD and not contribute to the inclusion; instead, they can play the role of cross-linker between the basic drug and the CD [37,38,39,40]. Consequently, a drug’s solubility can be improved, and the drug’s crystallinity is reduced, thereby accelerating its dissolution rate.

Although the total solubility achieved usually increases, in some cases a decrease in the complex stability constant was observed. The reason was that the higher drug solubility produced by an increased drug ionization in the presence of the acid resulted in less affinity for the apolar CD cavity [41,42].

Among the hydroxy acids, the citric, gluconic, tartaric, lactic and malic acids are reported to be promising ternary candidates due to their potential ability to interact with CD molecules by forming hydrogen bonds with their numerous hydroxyl groups [9,43].

Several studies on the ternary complexes of citric acid with CDs have been reported (Table 2, Figure 5). In the case of econazole nitrate, a salt commonly used in pharmaceutical formulations due to its great solubility, it was determined that the ternary system obtained by adding of an equimolar amount of citric acid to the SBEβCD binary complex produces a considerable increase in solubility [40]. Econazole nitrate’s solubility in the ternary system was 5.83 mg/mL; this value is higher than the sum of the maximum solubility obtained for each binary system, evidencing the synergy between SBEβCD and citric acid for the solubility improvement. In addition, this increase was reflected in a modified dissolution assay that simulated the oral cavity conditions, which showed an efficiency increase of 66-fold for the co-ground system compared with the drug alone. Moreover, the ternary complex showed significantly higher antimycotic efficacy against selected *Candida* strains than the drug alone [44]. Taking into account its superior properties, Jug et al. proposed the equimolar ternary system as a suitable product with therapeutic effectivity for the preparation of a novel mucoadhesive buccal formulation that has potential for drug delivery into the oral cavity, and it could improve on the existing therapy for oral candidosis [40,44]. Furthermore, ternary complexes of carvedilol with βCD and citric acid prepared at 1:2:2 molar ratios, respectively, by physical mixture, kneading and spray-drying, were reported. A notable solubility increase of 110-fold for the complex obtained by spray-drying was determined. In addition, this spray-dried complex showed greater efficiency for the complex formation than the kneaded complex. Tablets formulated with these complexes exhibited a significant improvement in dissolution compared with tablets containing the pure carvedilol, by achieving 100% release in 5 min. Therefore, this strategy is suitable for formulating mouth-dissolving tablets to overcome the first-pass metabolism of the drug and consequently improve its bioavailability [45]. Interestingly, Zhang et al. demonstrated the capacity of the clarithromycin–βCD–citric acid complex prepared by lyophilization, in a molar ratio of 1:1:1, to enhance the drug dissolution rate in basic media at pH 6.8. After oral administration of this ternary complex in beagle dogs, the authors suggested that the bioavailability of clarithromycin could be slightly improved by complexation with βCD and citric acid, due to the enhanced drug absorption in the duodenum, which was attributed to more of the drug being dissolved in the intestine and then available for immediate absorption [46].

In addition, multicomponent complexation via addition of hydroxy acids significantly improved cinnarizine’s solubility and the complexation efficiency of HPβCD. Among the ternary systems containing 1:1:1 molar ratios of drug–CD–hydroxy acid (citric acid or tartaric acid), the one prepared with tartaric acid via co-evaporation method increased the aqueous solubility of cinnarizine by 1223-fold [42]. The mechanisms through which the tartaric acid–CD ternary complexes improve the drug solubility have been extensively discussed in several reports. Aldawsari et al. reported that the combination oftartaric acid and HPβCD act as solubilizing agents for dapoxetine hydrochloride, a weakly basic drug with pH-dependent solubility that could limit its dissolution in physiologic neutral fluids. The pH-modifying role of tartaric acid enhanced solubility, which was explained on the basis of a stronger modulating effect on the pH of the microenvironment around the drug particles [47]. Complexation of iloperidone, an atypical antipsychotic drug, with HPβCD and tartaric acid, afforded a significant solubility increase of c.a. 17-fold (improved to a maximum of 200 μg/mL), and a better and faster dissolution profile, indicating the auxiliary roles of tartaric acid in increasing hydrophilicity and hydrogen bond formation of the ternary complex. A considerable reduction in the crystallinity of the drug was derived from intermolecular interactions with HPβCD and tartaric acid, which could be associated with the enhanced solubility given to iloperidone. This could result in enhanced bioavailability [38]. In addition, Yuvaraja et al. showed that the ternary system formed with HPβCD and tartaric acid was notably more effective at enhancing the solubility of carvedilol, a cardiovascular drug that is basic in nature, than the binary system with HPβCD. The solubility enhancement factor found was 340, which could be associated with the ionization process of carvedilol in the presence of tartaric acid. Moreover, this auxiliary agent increases the dissolution rate via control of the pH in the microenvironment [48].

Rakkaew et al. studied ternary complexes of haloperidol—an antipsychotic drug that is a weak base with very low aqueous solubility—with βCD and organic acids in different media. Tartaric, lactic and succinic acids were used to prepare the ternary complexes. In citrate buffer, pH 3, tartaric and lactic acids substantially increased drug solubility, which was attributed to unionized species of acid that interacted with haloperidol and CD by forming hydrogen bonds with the hydroxyl groups of βCD, which enhanced the solubilization capacity. In particular, the lactic acid produced the greatest binding strength and solubilization efficacy for the ternary complexes [37]. Afterwards, Chantasart and Rakkaew investigated different preparation methods of dry βCD-based ternary complexes of haloperidol and lactic acid for drug delivery purposes. The ternary complex prepared by freeze-drying showed a higher inclusion yield than those prepared by solvent evaporation and physical mixing. The ^1^H NMR studies confirmed that haloperidol interacts with the carboxyl and hydroxyl groups of lactic acid and CD inside and outside the cavity by hydrogen bonding or van der Waals interactions. In addition, drug amorphization was proven to occur due to the freeze-drying process, which significantly enhanced drug release. Complexes prepared using this technique could have superior haloperidol bioavailability, and may be useful in optimized carrier systems for nasal and transdermal delivery [49].

In another study, the ability of several acidic compounds to improve the formation of miconazole complexes with several CDs using supercritical carbon dioxide processing was reported [50]. The acids were able to promote drug inclusion, as shown by miconazole–HPγCD–tartaric acid/malic acid, and miconazole–HPβCD–tartaric acid, which achieved nearly complete inclusion. The hydroxypropylated CDs evidenced better results than native CDs. In particular, the presence of miconazole maleate salt was found for complexes containing maleic acid. Interestingly, these complexes were stable for up to one year. The complexes obtained with tartaric acid provided the maximum drug dissolution improvement. The best results were obtained with miconazole–HPγCD–tartaric acid in relation to miconazole alone. Subsequently, the relevant pharmacokinetic parameters of this ternary complex after oral administration were determined in pigs [51]. The complex formulation evidenced better absorption efficacy compared with capsules of miconazole alone, indicating higher relative bioavailability in an oral dosage form.

Analysis of the data in the literature showed that synergistic effects on the solubility of multicomponent complexes are obtained by adding appropriate organic acids as auxiliary agents. The effects of several multicomponent complexes on econazole, an antifungal base-type drug that is poorly soluble in water, were researched. Various CD (αCD, βCD, γCD and HPβCD) and hydroxy acid (tartaric, citric, gluconic, malic and lactic acids) combinations were studied. The multicomponent systems containing αCD were the best, as they produced the highest improvements in econazole solubility with almost all the various acid combinations. In particular, the system with αCD and malic acid in a 1:1:1 molar ratio was the most effective combination. It synergistically enhanced the aqueous drug solubility 2560-fold with respect to the drug itself and to 20-fold with respect to the binary complex [52]. Nuclear magnetic resonance (NMR) experiments and molecular modelling studies evidenced the complex formation with malic acid that involved interactions with the primary hydroxyl groups of αCD molecules [53]. Moreover, the synergic effect of the acids depended on the CD type in all the ternary systems examined. For example, citric acid showed 2480, 1540, 2460 and 2440-fold solubility enhancements with αCD, γCD, βCD and HPβCD, respectively, in comparison to econazole alone. Due to the drug solubility enhancements produced by the multicomponent systems, the ternary solid products showed faster dissolution rates than those of the drug alone, confirming their higher effectiveness [52]. Furthermore, Ribeiro et al. demonstrated the synergistic effect of tartaric acid and CD on enhancing the solubility of vinpocetine, which was related to a combined strategy of CD complexation and drug ionization. In fact, the formation of soluble multicomponent complexes increased the solubility of vinpocetine of 5 μg/mL to 5.80 mg/mL.; Moreover, an improvement in the complexation efficiency of βCD and SBEβCD was observed [41]. In addition, the combination of βCD and citric acid produced a synergistic effect on clarithromycin’s solubility: an almost 104-fold improved solubilization efficiency in relation to the intrinsic solubility of the pure drug [54].

Nevertheless, such ternary complexation was not always effective. For example, the ketoconazole–βCD–citric acid complex, in a 1:2:1 molar ratio, prepared via spray-drying method, showed a similar dissolution profile in a buffer solution at pH 5 to the binary complex ketoconazole–βCD prepared with the same molar ratio of βCD, although ketoconazole’s solubility was better in the ternary system than in the binary CD complex. This was attributed to the ionization of the drug in the dissolution medium [55].

## 4. Organic Bases as Auxiliary Agents

Organic bases, whose structural diversity predetermines their magnitudes of effect upon drug–CD solution interactions, are being employed as tertiary components in ternary complexes, and they are used mainly when the drugs are acids. In these systems, the prominent role of electrostatic forces in the general interaction should be considered. Although the CD complexes obtained from non-ionized drugs have greater stability compared with their anionic analogues [10], the total solubility achieved and other properties of the drug such as chemical stability and bioavailability usually improve.

In the study carried out by Redenti et al. [10] on salt formation and concomitant complexation with CD to improve oral performance for acidic drugs, hydroxylamines were used with non-steroidal anti-inflammatory drugs. The aqueous solubility of piroxicam, a poorly soluble derivative of benzothiazine, is significantly increased by simultaneous salt formation with sodium, potassium and ammonium, and complexation with ɣCD, achieving 4–7-fold and 2–4-fold solubility enhancements with respect to the drug itself and the binary complex, respectively [56].Simultaneous salt formation and complexation with βCD or HPβCD markedly increases the solubility of indomethacin, an arylacetic derivative, at pH 6; enhancements of 15- and 6-fold, respectively, were detected with respect to the corresponding binary complexes. These systems also led to better in vitro release in a simulated gastric environment [57].

Several therapeutic drug families include acid groups in their structures, such as carboxylic acid, phenol and sulfonamide. For example, nonsteroidal anti-inflammatory agents such as flurbiprofen and diclofenac; diuretics that are carbonic anhydrase inhibitors, such as furosemide and acetazolamide, which is also used for the treatment of glaucoma, among other pathologies; and antibacterials such as sulfisoxazole and norfloxacin. Almost all exhibit several disadvantages, including low aqueous solubility, limited bioavailability and chemical instability [10], which often makes it difficult to fully take advantage of their therapeutic properties. In particular, for drugs belonging to these categories, for which rapid onset of therapeutic action is required, such as analgesics, and whose absorption is not hampered by poor permeability through membranes, poor solubility or slow dissolution could delay the desired effects. In addition, slow dissolution of anti-inflammatories can also increase local side effects associated with medications (e.g., gastric irritation) [58]. Sulfisoxazole is a weak acidic antibacterial that, like many sulphonamides, exhibits poor aqueous solubility, offering difficulties in pharmaceutical formulation. To overcome this drawback, a promising alternative is complex formation with CDs, which may allow an increase in the aqueous solubility of sulfisoxazole. However, if the complexation efficiency is not very high, relatively large amounts of CDs must be used to complex small amounts of the drug, in order to achieve the required solubility increase. Granero et al. demonstrated that the aqueous solubility of sulfisoxazole was significantly improved by the formation of multicomponent complexes with HPβCD and a basic substance such as triethanolamine (TEA) [59]. In addition, the sulfisoxazole–TEA–HPβCD complexes have been studied both in solution by phase-solubility analysis and in solid state using differential scanning calorimetry (DSC), thermogravimetric analysis (TA), infrared spectroscopy (IR) and dissolution assays. The authors observed that the dissolution profiles of the lyophilized inclusion complexes showed faster dissolution rates compared than the prepared physical mixtures and the pure drug; however, the improvement was better with the ternary system.

Taking into account that nonsteroidal anti-inflammatories exhibit slight water solubility, some authors have suggested their inclusion in CD complexes to revert this problem. Accordingly, ternary CD complexes with ethanolamines were obtained by Maitre et al. [60] to improve the solubility of flurbiprofen, since it is one of the most potent inhibitors of platelet aggregation currently available. However, because flurbiprofen is conventionally administered orally, it causes a number of gastrointestinal disorders that become an issue in terms of side effects. For this reason, and taking into account that some researchers have studied the effects of CDs on transdermal permeation, these authors developed a system that allowed not only an increase in the solubility of flurbiprofen, but also application of the drug through the skin. Since the efficiency of complexation is not frequently very high, it can be improved through the use of a proper third component. In addition, they also found that the formation of salts with monoethanolamine (MEA) or diethanolamine (DEA) improved the percutaneous absorption of piroxicam, another anti-inflammatory drug [61]. Considering the above concepts, for another study the authors decided to use the hydroxypropyl derivative of βCD, HPβCD, due to its higher aqueous solubility, and three ethanolamines (Figure 6) (MEA, DEA and TEA). In addition, they highlighted that as of the time of publication, no studies had been reported that addressed the influence of joining CD complexes of lipophilic drugs with ethanolamines, thereby simultaneously exploiting CD and ethanolamines’ solubilizing power to improve the dermal absorption, favoring delivery through the skin. Solubility studies of flurbiprofen and each alkanolamine showed that TEA and DEA were effective at improving the solubility through a synergistic effect. On the contrary, ternary systems with MEA and HPβCD did not show a positive effect. The solid complexes, prepared through the freeze-drying method, were characterized by IR, DSC and TA. From the results provided by these studies, the salt formation of flurbiprofen was corroborated by interactions with alkanolamines and complexation with HPβCD. In dissolution studies, it was found that a strong increase in the dissolution profile could be observed with the ternary systems, compared with the pure drug and the binary complexes. Flurbiprofen–HPβCD–MEA showed the best profile. In view of the results obtained in the first stage of this work, the authors studied the effect that complexation with MEA and HPβCD could have with the addition of binary cosolvent mixtures that are enhancers of the permeation, such as isopropyl myristate/ethanol and isopropyl myristate/propylene glycol, on the solubility of flurbiprofen. The mixing of HPβCD with the binary cosolvent systems, consisting of isopropyl myristate and propylene glycol, significantly enhanced the skin permeation for the flurbiprofen–MEA complex after 12 h of application. This combination gave the best permeation profile for this complex among all the formulations tested. In this system, HPβCD appears to exert a cooperative effect toward enhancing the drug complex’s permeation.

Garnero and Longhi [62] have studied the interactions of ascorbic acid with HPβCD and TEA, separately and in combination. The first objective established for this work was to verify the interactions that could exist among ascorbic acid, HPβCD and TEA. For these investigations, physicochemical methods such as DSC, FTIR, ^1^H NMR and ^13^C NMR spectroscopies were used to analyze the binary and multicomponent complexes and physical mixtures. In addition to the above, the authors investigated the formation of HPβCD aggregates, and whether the systems obtained improved the stability of ascorbic acid in solution, though which the values of the stability constants of the complexes and the degradation rates of ascorbic acid were obtained from degradation studies. Comparisons of the results obtained by DSC and FTIR of the systems prepared by freeze-drying with those obtained by physical mixing confirmed interactions between the components. The changes observed in the DSC thermograms and in the IR spectra are undoubtedly clear evidence of interactions between ascorbic acid and the other two compounds, suggesting the formation of a complex. By the NMR studies, it was possible to propose the formation of an association complex between ascorbic acid and TEA, which then formed a ternary system through interactions with the hydroxyl groups on the external surface of the CD. The evidence on the formation of aggregates was obtained from the changes experienced in the chemical shifts of the protons towards high fields, from the graphs produced for the pure HPβCD solution, the HPβCD solution containing ascorbic acid and the HPβCD solution containing ascorbic acid and TEA, as a function of the inverse of the total concentration of HPβCD. Through these findings, it was proposed that the formation of aggregates of HPβCD is promoted by ascorbic acid. As ascorbic acid degrades rapidly in aqueous solutions, its effects on stability when complexed with HPβCD and TEA, either separately or in combination, were studied by obtaining graphs of the observed rate constants as a function of the ligand concentration. The conclusions of these studies were that there is an increase in the stability of ascorbic acid after the addition of TEA, and that the multicomponent complex causes even greater stabilization than HPβCD alone. In addition, an approximately 17-fold increase in the stability of ascorbic acid was evidenced in the presence of both ligands in an aqueous solution.

Acetazolamide, a drug used orally to lower intraocular pressure in glaucoma patients, is a diuretic and a carbonic anhydrase inhibitor. Large doses of the drug must be used to obtain the desired reduction in intraocular pressure, which usually produces systemic side effects, the most frequent being diuresis and metabolic acidosis. Therefore, topical administration of acetazolamide is preferred over systemic administration. The two major problems that hinder the topical effectiveness of acetazolamide are its poor aqueous solubility and its low corneal permeability. Therefore, taking into account the positive results previously obtained with sulfisoxazole, Granero et al. [63] decided to prepare ternary inclusion compounds of acetazolamide, HPβCD and TEA, intended for topical ocular administration. A significant increase in the solubilizing power of HPβCD was obtained by the simultaneous complexation and salt formation with TEA. The scarcely water-soluble acetazolamide, when forming a ternary system of drug–HPβCD–TEA, is more available in aqueous HPβCD solutions, which should make the complex more effective for drug delivery to the cornea’s surface. Furthermore, the top results in release studies with formulations containing HPβCD and TEA highlighted TEA as an effective additive for the corneal transport of acetazolamide in aqueous eye drops. In addition, the pH of the formulation containing the ternary complex was within physiological pH values. The ternary complexes prepared by the freeze-drying method were analyzed by means of IR, and according to the changes in the spectra, it was suggested that the carbonamido group of the drug is involved in the inclusion process, which is reflected by the shifts in the C=O vibrational peak. The in vitro release studies of acetazolamide for all the formulations were carried out using the membrane diffusion technique. They indicated that acetazolamide–HPβCD–TEA was significantly better than the others. When comparing the release data of acetazolamide in pure water with those of the binary inclusion complex and its corresponding physical mixture with acetazolamide and HPβCD, it is clear that the acetazolamide release was faster from formulations containing the binary systems with HPβCD. Besides, the drug release from the inclusion complex was more complete than that of its corresponding physical mixture. It was found that the results of the in vitro release studies were in agreement with those obtained from the solubility ones, in which the ternary complex acetazolamide–HPβCD–TEA showed the highest drug solubility. Perhaps the difference in release rate could be attributed to the changes in solubility shown by the different formulations of acetazolamide. Furthermore, significantly better results with formulations containing HPβCD and TEA in the release studies established TEA as an effective additive for the corneal transport of ACZ in aqueous eye drops. Since the eye tolerates a narrow range of pH, it is important to note that formulations of binary and ternary inclusion complexes of acetazolamide with HPβCD and/or TEA maintain the pH within the physiological pH range of 6.2–8.4, thereby respecting the pH range tolerated by the eye. In addition, both binary and ternary acetazolamide systems with HPβCD and TEA were shown to allow excellent drug stability over several days.

Due to the good results obtained in the previous study, the authors decided to continue with the investigations regarding acetazolamide complexes with HPβCD alone or with TEA by studying crystalline properties, dissolution and the effect on intraocular pressure [64]. They found that the preparation conditions could vary the crystal structure of acetazolamide powder. The particle morphology and the polymorphic form were changed after freeze-drying, whereby the starting acetazolamide was converted to pure form A. The solid products obtained by freeze-drying, and the physical mixtures prepared for comparative purposes using an agate mortar, were characterized by DSC, TA, X-ray powder diffraction (XRPD), scanning electron microscopy (SEM) and FTIR, and the obtained results suggested the formation of inclusion complexes of acetazolamide with HPβCD alone or with TEA, by the freeze-drying method, and the conversion of the drug into the amorphous state. In addition, binary and ternary systems of acetazolamide obtained by freeze-drying exhibited satisfactory acetazolamide dissolution rates. The effects of the drug and its complexes with HPβCD alone or with TEA regarding the drop in the intraocular pressure (IOP) in normotensive rabbits were investigated. With these binary and ternary lyophilized acetazolamide systems, the maximum IOP-lowering effect obtained with these binary and ternary lyophilized acetazolamide systems occurred at around 90 min, whereas a longer maximum IOP-lowering effect was shown by the ternary complex acetazolamide-TEA–HPβCD than the binary system. These results agree with those obtained in dissolution studies, in which the ternary system showed longer dissolution times in relation to the lyophilized binary one. Considering that large oral doses of acetazolamide lower IOP, but usually lead to a multitude of systemic side effects, including gastrointestinal upset, researchers [65] decided to evaluate the effect of acetazolamide on the histological structure of rat duodenal mucosa. They studied the protective effect that the acetazolamide–HPβCD complex could exert, with and without the addition of a third compound, on the gut epithelial layer, through histological and ultrastructural examinations of sections of rat duodenum exposed to acetazolamide or its formulations. Furthermore, the passage of acetazolamide and its binary or ternary complexes through the duodenal mucosa was analyzed using the single-pass intestinal perfusion method in rats. It was determined that acetazolamide modifies intestinal permeability and injures the small intestine of the rat, and that acetazolamide complexes significantly mitigate the damage caused by the drug on the mucosa, which was proven by microscopic studies. In addition, it was found that the ternary complexes of acetazolamide with HPβCD in combination with TEA or calcium ions markedly improved the apparent permeability of acetazolamide through the duodenal epithelium, alongside the additional property of maintaining the integrity of the intestinal epithelium after drug administration. These excellent results of ternary acetazolamide systems with HPβCD, in the presence of TEA or calcium ions, can be attributed to the rapid release of acetazolamide into the underlying epithelium, thereby enhancing the amount of drug available for absorption. The authors also postulated that the obtaining of ionic pairs competes with the complex formation and could shift the equilibrium towards the dissociation of the complex, decreasing its stability.

Like flurbiprofen, diclofenac is also characterized by low aqueous solubility, poor dissolution and gastrolesive actions, whereby the complexation with MβCD and MEA was investigated in order to determine if this system alleviates these undesirable effects. Its influence on the percutaneous absorption of diclofenac is being studied [66]. Transdermal delivery of diclofenac would be advantageous, since it would avoid hepatic first-pass metabolism and considerable gastrointestinal disturbances. In solution the complexes were studied by phase-solubility analysis and ^1^H NMR spectroscopy, and these results indicate that 1:1 inclusion stoichiometry was present, and the ^2^D ROESY NMR experiment showed that in the inclusion complex of diclofenac and MβCD, the dichlorophenyl part of the drug was included in the MβCD cavity. Moreover, the solid complex was prepared by the freeze-drying method and analyzed using DSC and TA. These studies allowed obtaining more information on possible drug–ethanolamine solid-state interactions. The facts observed in the thermograms, such as disappearance, shifting, decreasing or widening of the peaks, could be indicating a complexation of the diclofenac–MEA salt into the cavity of MβCD. In addition to the above, the cumulative permeation of diclofenac from different formulations through human skin was investigated, and it could be determined that a drug permeation enhancement of 67-fold was obtained at 30 h, with the ternary diclofenac–MβCD–MEA complex. This good result was attributed to the fact that MEA prevents complexation, allowing greater availability of the free drug, and thus promoting the skin absorption of diclofenac.

Benznidazole, a drug mainly used to treat Chagas disease, is classified as class II in the BCS due to its low solubility and high rate of penetration through biological barriers. Due to these properties, it would be beneficial to increase the solubility of the drug through the use of technological alternatives, which increase its efficacy. Due to this, Nunez de Melo et al. [67] selected complexation with βCD for this purpose, among the various techniques available to improve the solubility of nonpolar drugs in aqueous vehicles. Multicomponent complexes of benznidazole and βCD were prepared in the presence of hydrophilic polymer and TEA. By phase solubility-diagrams, in the systems with alkanolamine at different pH values, the occurrence of soluble complexes with 1:1 stoichiometry was determined, but the solubility in the ternary system was lower than that obtained for the binary with βCD, with which it was possible to increase by 2-fold the solubility of the pure drug. The authors attribute this fact to a co-solvency effect or competition for the cavity of βCD. Through molecular dynamic simulations, it was shown that TEA acts by hindering the formation of the benznidazole–βCD complex in accordance with the experimental results, which showed lower solubility of benznidazole due to such competitive phenomena. In addition, through the ^1^H NMR studies they carried out, it was found that all benznidazole protons have greater protective effects with the addition of TEA to the binary system with βCD, which implies a strong interaction between benznidazole and TEA, thereby confirming the co-solvency effect.

In another study, the same group [68] published the preparation of spray-dried complexes of benznidazole with βCD and TEA, which were carefully controlled using FTIR, XRPD and SEM images. The results obtained allowed verifying the obtaining of systems that involve the three components, but on the other hand, the presence of the TEA did not significantly change the benznidazole dissolution, compared with the binary complex. In previous scientific works, it was possible to verify the synergistic effect that βCD and TEA exert on molecules that are not very soluble in water and are acidic, which does not seem to happen for benznidazole, due to its neutral nature and non-ionization capacity. In contrast, TEA appears to disturb the insertion of benznidazole into the hydrophobic cavity of βCD, which may explain the failure of TEA to increase the aqueous solubility of benznidazole:βCD.

Methotrexate is an anticancer drug that is also used to treat “psoriasis,” which is a chronic inflammatory disease that causes skin lesions. However, due to its short half-life (1.5 to 3.5 h), frequent administration of high doses of the drug is required, which is associated with the appearance of side effects. Therefore, an alternative to increase the effectiveness of methotrexate for the treatment of psoriasis would be its application directly to the lesions with the consequent permeation of the drug through the skin, which would restrict the possible undesirable effects. Nevertheless, the topical application of methotrexate has certain limitations. Due to this, Barbosa et al. [69] aimed to develop nanotechnological drug delivery systems based on CD in an attempt to solve the drawbacks exhibited by this drug permeation. The selected strategy was to obtain ternary complexes of βCD with TEA, taking into account the capacity of alkanolamine to enhance the inclusion of the host drugs by CD. Additionally, especially in relation to ternary complexes for topical application, TEA proved to be an efficient complement with which to enhance the corneal release of acetazolamide, thereby improving the in vivo efficacy of the drug and decreasing its unwished consequences. The correlations in an aqueous medium were investigated by molecular modeling, phase solubility diagrams and NMR studies. The solid systems were prepared by the freeze-drying method, and were characterized using SEM, IR, DSC, TA, C,N,H elementary analysis, XRPD and in vitro drug dissolution studies. The linear phase diagram obtained suggested the production of soluble complexes with 1:1 stoichiometry. The maximum βCD concentration studied increased the apparent methotrexate aqueous solubility about 15-fold. Besides, with the addition of TEA to the binary samples, a synergistic effect was observed, with increments in the drug solubility of 30-fold. In the ^1^H NMR studies, the ROESY spectrum obtained for binary and ternary complexes showed interactions between the aromatic protons of methotrexate and the protons of the internal cavity of β-CD, indicating that the alicyclic aromatic ring of the drug is deeply inserted in the CD cavity, in both binary and multicomponent complexes. In the molecular modeling study, it was shown that TEA significantly increased the affinity of methotrexate for β-CD through an electrostatic interaction that was not observed for the binary complex. Unlike in other studies, TEA did not compete with the CD cavity, but it stabilized the methotrexate–βCD interaction. For methotrexate–βCD–TEA freeze-dried complexes, the IR results proved that the drug–CD interactions identified by NMR studies remained in solid phase. Thermal and structural studies also pointed out this fact, and highlighted the increase in thermal stability of the drug. The impacts of the ternary complex on the objectives were further confirmed by the results obtained in the in vitro dissolution studies. In vitro drug release studies that were performed for pure methotrexate and for binary and ternary freeze-dried complexes with the objective of evaluating the effect of complexation on the drug dissolution rate revealed much faster dissolution by the drug in both in binary and ternary freeze-dried complexes. The dissolution performance of the ternary complex was the fastest, demonstrating the success of the lyophilized ternary complex.

## 5. Water-Soluble Polymers as Auxiliary Agents

Water-soluble polymers are known to interact with the outer surfaces of CD and drug–CD complexes, forming aggregates or co-complexes that show values of stability constants constant higher than those of binary drug–CD systems [70]. They play a role in stabilization, preventing aggregations of complexes and several types of particulate in pharmaceutical systems. They can also increase the solubility of complexes and decrease CD mobility by changing the hydration properties of CD molecules [71]. In this review, the use of different water-soluble polymers as third components in multicomponent CD complexes is analyzed.

### 5.1. Polyvinylpyrrolidone

Polyvinylpyrrolidone (PVP) can be used as an aqueous solution or a powder that can be dissolved in various organic solvents and water. It has excellent physiological compatibility, complexing ability, film-forming capability and hygroscopic capacity. PVP can be used in cosmetics as a viscosity-enhancing agent, a film agent, an adhesive and a lubricant. In the pharmaceutical field, PVP-K30 (Figure 7A) is an excellent excipient used as a dissolution assistant for injections, a binder for tablets, a dispersant for liquids, a flow assistant for capsules, a stabilizer for enzymes and heat sensitive drugs, an antitoxic element and lubricator for eye drugs and a coprecipitant for poorly soluble drugs [72]. Efavirenz is the most widely used drug for the antiretroviral treatment of acquired immunodeficiency syndrome. However, it has low solubility and it does not exhibit appropriate bioavailability. It is classified according to the BCS as a class II drug, with high permeability but low aqueous solubility (~3–9 µg/mL) [73], which interferes with its therapeutic action. Among several drug delivery systems, the multicomponent systems with low-concentration CDs and hydrophilic polymers are the most promising alternatives for increasing the aqueous solubility of drugs [74,75]. Efavirenz–MβCD–PVP-K30 with 1% PVP-K30 promoted the best increase in the solubility and delivery of efavirenz, when the dissolution profiles were analyzed. More than 80% of the total efavirenz was delivered in 30 min, compared with less than 25% by efavirenz alone. That is, the PVP-K30 represented a co-complexing agent that increased the hydrophilic potential of MβCD [67,76]. The use of the kneading method to obtain the multicomponent solid-state system permitted the formation of a non-crystalline, uniform particle, which increased the dissolution rate of the drug, and provided an increase in the stability of the drug, as demonstrated by thermal analysis. DSC analysis suggested that the presence of MβCD and PVPK-30 delays the efavirenz melting process, leading to both a change in its peak temperature and widening of the peak, with a consequent increase in the energy involved. Consequently, it is suggested that the system gives stability to efavirenz. Moreover, TA showed that the kneaded product loses less mass, demonstrating that the product obtained by kneading is more stable [75] due to strong electrostatic interactions between PVP-K30 and MβCD, as shown in the IR spectrum.

PVP-K30 was also used in a multicomponent complex with βCD and cefuroxime axetil, to improve the latter’s pharmaceutical characteristics, via the kneading method [77]. Cefuroxime axetil is a prodrug of cefuroxime cephalosporin, which is used for pharyngitis, respiratory tract infections, acute bacterial otitis, simple skin infections and urinary tract infections. It is a BCS class II drug, so it has less aqueous solubility with an oral bioavailability of 37% on an empty stomach, which increases to 52% if taken after meals. In this multicomponent complex of cefuroxime axetil with βCD, PVP-K30 can act as a ternary component to improve its pharmaceutical characteristics. The multicomponent complex leads to a better dissolution profile than the βCD–cefuroxime axetil complex: >85% cefuroxime axetil release within 30 min. This release is much more effective due to the hydrophilic nature of PVP-K30, which radically improves the phase solubility parameters, such as stability constant and complexation efficiency, while interacting simultaneously with cefuroxime axetil through electrostatic interactions and salt formation, and βCD via H-bonding. The in vitro taste-masking study also revealed that the multicomponent complex was able to mask the bitter taste of the cefuroxime axetil [78]. Cefuroxime axetil’s physicochemical characteristics were also improved via the dual mechanism of HPβCD complexation in the presence of a water-soluble polymer and simultaneous particle size reduction using spray-drying technology [78]. The polymers are believed to enhance the stability constants of drug–CD complexes, the interactions with drug–CD complexes and the interactions with the outer surfaces of CDs [79]. The addition of water-soluble polymers supplies an advantage through the complexation efficiency of HPβCD by establishing molecular interactions based on the solubilization of the amorphous form of cefuroxime axetil, such as Van der Waals dispersion forces, hydrophobic bonds or hydrogen bonds in the presence of PVP [80]. The particle size revealed that the binary and multicomponent complexes were micro-sized. This led to a rise in the specific surface area, and therefore, to faster dissolution for cefuroxime axetil particles. Therefore, an increase in bioavailability would be expected.

Solid-state ibuprofen manipulation continues to be a challenge for researchers because of its low glass transition temperature and its tendency to recrystallize at room temperature. For this reason, the formation of a water-soluble and stable multicomponent system ibuprofen–βCD–PVP-K30 in a solid state was evaluated, in order to simultaneously improve the physical stability and the dissolution of ibuprofen [81]. This system was obtained through the milling technique, which represents an economic, eco-friendly and simple process. The milling technique has succeeded at improving the bioavailability and the solubility of numerous poorly water-soluble drugs. The formation of such a multicomponent system denotes a new alternative to improving the bioavailability and solubility of ibuprofen, which preserves its stability in an amorphous state. This system shows a substantial enhancement in dissolution rate compared with pure ibuprofen. After 0.5 min, the amount of ibuprofen released by the multicomponent system was practically quadrupled compared to the drug alone and the binary systems ibuprofen–βCD and ibuprofen–PVP-K30. In this system, PVP-K30 may partially or totally coat the ibuprofen–βCD complex, leading to the formation of a stable and water-soluble amorphous system. The physical stability of amorphous ibuprofen (at RH: 75%/T = 40 °C for 6 months) can be achieved by forming a multicomponent system in the ratio 1:1:0.5 *w*/*w* obtained by milling the drug into solid state at 25 °C. IR and NMR spectroscopies have suggested the presence of electrostatic interactions and intermolecular H-bonds between ibuprofen molecules and carriers. XRD, ^1^H-NMR shifts and SEM results have clearly revealed the formation of amorphous complexes coated by PVP-K30 films. This represents an advantage in terms of ibuprofen release in comparison with ibuprofen–βCD and ibuprofen–PVP-K30 in a 1:1 *w*/*w* ratio [82].

### 5.2. Chitosan

Chitosan (CH) (Figure 7B) is a biopolymer of aminopolysaccharides composed of randomly distributed units of β-(1-4) D-glucosamine (deacetylated units) and *N*-acetyl-D-glucosamine (acetylated unit). CH is a biocompatible, antibacterial and environmentally friendly polyelectrolyte with a variety of applications, including water treatment, chromatography, additives for cosmetics, biodegradable films, biomedical devices and microcapsule implants for controlled release in drug delivery [83]. CH has been widely reported for its ability to increase the permeability of drugs [84] and for its availability as a carrier system for drug targeting [85]. Trimethyl chitosan (TMC) is a derivative of CH especially known for its abilities to increase drug bioadhesiveness and biocompatibility [86], and permeation and bioavailability through the intestinal mucosa [87]. In the case of modafinil, the complexation efficiency obtained by the use of TMC is better compared with the binary system from the manufacturing point of view because TMC can be produced via easier product manufacturing compared with the complex containing CD alone. The aggregation of CD derivatives by TMC and the creation of a microenvironment with high solvation power could be the mechanism behind the ability of TMC to impact the modafinil solubility [88]. The phase solubility study and its correlation with the DSC study proposed that the high dissolution nature of the metastable polymorphic form, which has not been converted into a stable but poorly soluble polymorphic crystalline [89] form, could be the reason for the solubility enhancement of TMC-mediated multicomponent modafinil complex. On the contrary, unmodified CH-based multicomponent complex indicated absolute conversion of modafinil crystalline form to amorphous form during freeze-drying.

### 5.3. Hydroxypropylmethylcellulose

Hydroxypropylmethylcellulose (HPMC) (Figure 7C) is a water-soluble semisynthetic derivative of cellulose substituted with methoxy and hydroxypropyl groups [90]. Diosmin, a drug with poor water-solubility which limits its therapeutic applicability, possesses strong antioxidant activity, along with other interesting effects, including anti-inflammatory, anti-cancer and anti-ulcer activities [91]. Therefore, binary and multicomponent complexes with CD and CD plus HPMC or PEG were assayed by Anwer et al. [92]. The stability constants determined for the diosmin–βCD system and diosmin–βCD–HPMC system were 151 and 186 M^−1^, respectively, suggesting relatively strong interactions between βCD and the diosmin. These interactions were established through intermolecular forces between the βCD cavity and diosmin, resulting in the inclusion of diosmin within the βCD cavity. A significant rise in solubility was achieved through the addition of 1% *w*/*w* of HPMC in the presence of βCD.

Norfloxacin is a poorly soluble drug, classified as a BCS class IV agent. The effect of HPMC and βCD on norfloxacin was investigated by the solvent evaporation method to prepare the inclusion complex. Through phase solubility studies, the stability constant value was increased from 103.49 M^−1^ for β-CD to 253.34 or 307.49 M^−1^ when 2.5% (*w*/*v*) or 5% (*w*/*v*) of HPMC were added. These values showed the formation of a multicomponent complex comparatively more stable than the binary one. The complexation with βCD significantly enhanced the dissolution rate of norfloxacin in comparison with pure norfloxacin. However, HPMC did not show a significant increase in norfloxacin release [93].

### 5.4. Hyaluronic Acid

Celecoxib is a nonsteroidal anti-inflammatory drug that has inhibitory effects on vascular endothelial growth factor, as it demonstrated in numerous anticancer studies with diverse cell types through the inhibition of the COX-2 enzyme, and it has anti-proliferative and anti-angiogenic effects on several cell types, including endothelial cells [94,95]. It is used in macular degeneration and diabetic retinopathy. However, the oral dose required to obtain a therapeutic effect is very high, which results in cardiovascular problems and adverse effects [96]. Intravitreal injections can offer a high concentration of celecoxib to the retina, but can potentially produce retinal toxicity. Moreover, recurrent intravitreal injections have been associated with complications such as endophthalmitis and retinal detachment [97]. Consequently, there is a need to develop non-invasive methods for effective retinal drug delivery. As stated above, Jansook et al. [98] developed binary and multicomponent complexes of celecoxib (containing RMβCD and γCD as solubilizers, and HPMC and hyaluronic acid–HA, Figure 7D, as mucoadhesive polymers) to try to optimize these unfavorable properties. HA is a non-sulphated glycosaminoglycan and is composed of repetitions of polymeric disaccharides of D-glucuronic acid and *N*-acetyl-D-glucosamine linked by glucuronidic β (1-3) bonds. It forms specific and stable tertiary structures in aqueous solutions [99]. Celecoxib–CD suspensions with 0.5% *w/v* HA or HPMC provide good mucoadhesive properties that were believed to be able to raise celecoxib’s residence time, resulting in enhanced ocular drug bioavailability. Different systems were assayed, and the microsuspension eye drops with multicomponent complex celecoxib–RMβCD–HA aggregates exhibited relatively good mucoadhesion; provided the maximum drug flux through the simulated vitreous humor, through the semipermeable membrane and through scleral tissues; and had cytocompatibility with the human retinal pigment epithelial (RPE) cell line [98].

### 5.5. Polyethylene Glycol

Polyethylene glycol (PEG) (Figure 7E) is chemically a petroleum-based polyether that is much used in medicine and industrial manufacturing. PEG is also called polyethylene oxide or polyoxyethylene, according to its molecular weight. PEG’s structure is generally expressed as H−(O−CH_2_−CH_2_)n−OH. For example, diosmine complexation with PEG 6000 increased the stability constant value from 151 M^−1^ for the binary complex to 236 M^−1^ for the diosmine–βCD–PEG 6000 system. This showed that a significant increase in solubility was reached by the addition of 1% *w*/*w* of PEG 6000 in the company of βCD. According to dissolution studies, the dissolution of diosmine was found to be 14% or 21% at pH 1.2 or 7.4 after 60 min, respectively. Release of kneaded diosmine–βCD complex was increased to 29% (2-fold) and 56% (2.6-fold) at pHs 1.2 and 7.4, respectively. However, significant rises in dissolution of 35% (2.4-fold) at pH 1.2 and 73% (3.4 fold) at pH 7.4 were observed for the diosmine–βCD–PEG 6000 complex, which may be attributed to the amorphous nature of the solid that was confirmed by XRPD studies. Moreover, this increase in solubility produced the highest radical scavenging activity (75% at 100 μg/mL) for the diosmine–βCD–PEG 6000 complex [92].

### 5.6. Poloxamer

Poloxamer (P) is a non-ionic polymer, a block copolymer made of poly(oxy ethylene) and poly(oxy propylene) (Figure 7F). Some previous works have shown that the addition of P to CD leads to the formation of drug nanocarriers and colloidal particle systems [100]. The drug’s incorporation into P micelles can improve chemical stability and drug solubility [101] and also regulate the cell accumulation acting on efflux pumps and biodistribution [102].

One of the main components of the *Centella asiatica* is asiaticoside, which has become interesting today for its wound healing mechanism. Asiaticoside was found to significantly improve the rate of wound healing and enhance the tensile strength of wound tissues [103]. Due to its poor aqueous solubility, permeation through the biological membranes is difficult, leading to low bioavailability. Interestingly, the additive effect of SBEβCD solubilization of asiaticoside was observed with P407. The enhancement of solubilization with the addition of P407 to asiaticoside–SBEβCD was approximately 12%. This is high enough to allow less SBEβCD to be used in the final formulation. In addition, a slight dissolution retardant effect was observed for the asiaticoside–SBEβCD–P407 complex. When the multicomponent complexes came into contact with the dissolution medium, the polymers swelled and formed a hydrophilic layer, resulting in a slow diffusion rate for the drug. However, asiaticoside’s release rate from asiaticoside–SBEβCD–P407 was increased up to 93% in 120 min. The stability constant of the asiaticoside–SBEβCD complex was slightly higher than that of the multicomponent complex. There was only a slight enhancement of dissolution efficiency at 120 min for the multicomponent complex. This successful dissolution rate might be due to the decrease in crystallinity of the asiaticoside during the freeze-drying process [104].

β-lapachone is a 1,2-orthonaphthoquinone derived from the woof of the lapacho tree. β-lapachone provoked cell death in human cancer cells [105] through a mechanism that requested the presence of the NAD (P)H–quinone oxidoreductase-1. β-lapachone administration increases the generation of reactive oxygen species in cells or tissues with the presence of NQO1, causing extensive DNA damage. Moreover, β-lapachone causes losses of NAD+ and ATP pools, which then inhibits DNA repair and accelerates cell death [106]. NQO1 is overexpressed in breast tumors with respect to normal adjacent tissue, which makes β-lapachone a potential selective anti-tumoral drug for breast cancer [107]. However, the low aqueous solubility (0.038 mg/mL) and unspecific distribution of β-lapachone limit its use in clinical assays [108]. Seoane et al. [109] prepared a multicomponent system of β-lapachone in an optimal random MβCD–poloxamer 407 mixture. Then, its anti-tumor effects on proliferation, apoptosis, cell cycle, tumor growth and DNA damage were evaluated in vitro and in vivo using MCF-7 cells, human breast adenocarcinoma cells and immunodeficient mice. This multicomponent system is fluid at room temperature, gels at over 29 °C, and delivers a significant amount of β-lapachone, thereby facilitating intratumoral release, in situ gelation and the formation of a depot for time-release. When the β-lapachone multicomponent system was administered to MCF-7 cells, no changes in cell cycle were induced, despite producing increases in apoptosis and DNA damage. Additionally, intratumoral injection of the system into a mouse xenograft tumor model significantly decreased tumor volume while increasing apoptosis and DNA damage without evident toxicity to the liver or kidneys.

## 6. Conclusions

CDs are excipients widely used by the pharmaceutical industry that are incorporated in many pharmaceutical formulations marketed in several regions of the world. In recent decades, the development of multicomponent systems has increased the potential for CDs to not only to enhance drug solubility, dissolution and bioavailability, but to also allow the modulation of other drug properties, such as the stability and biological activity. The multicomponent complexes of CD with organic acids or bases, amino acids and polymers have proven to be very useful for improving and controlling properties of ionizable, weakly ionizable and non-ionizable drugs. In general, the most suitable auxiliary agent and the most appropriate CD should be selected by considering the particular drug’s physicochemical properties and the characteristics to optimize. For example, in the case of acidic drugs, the basic auxiliary agent TEA has been demonstrated to significantly enhance both the solubility and the permeability. Regarding using amino acids as auxiliary agents, arginine is the most used compound, since in general it shows a synergistic effect with the CD toward enhancing drug solubility. In summary, our analysis of the available literature showed the interest in multicomponent CD complexes as approaches with which to improve the therapeutic efficacy of drugs.

## Figures and Tables

**Figure 1 pharmaceutics-13-01099-f001:**
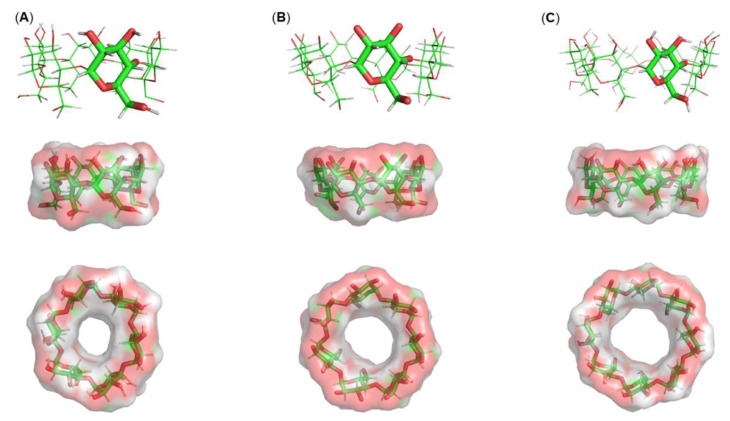
Molecular structures and three-dimensional side and top views of: (**A**) αCD, (**B**) βCD and (**C**) γCD. The surfaces shown represent the solvent-accessible surface areas (SASAs) in regard to water. Colors by atom-type indicate the regions of the molecule that are accessible (red: oxygen; grey: carbon). SASAs were calculated using cyclodextrin structures deposited in the Cambridge Crystallographic Data Centre under 1100537, 1107195 and 1126611 deposition numbers for αCD, βCD and γCD, respectively. The image was prepared using Pymol v.2.1 and in-house developed Python scripts.

**Figure 2 pharmaceutics-13-01099-f002:**
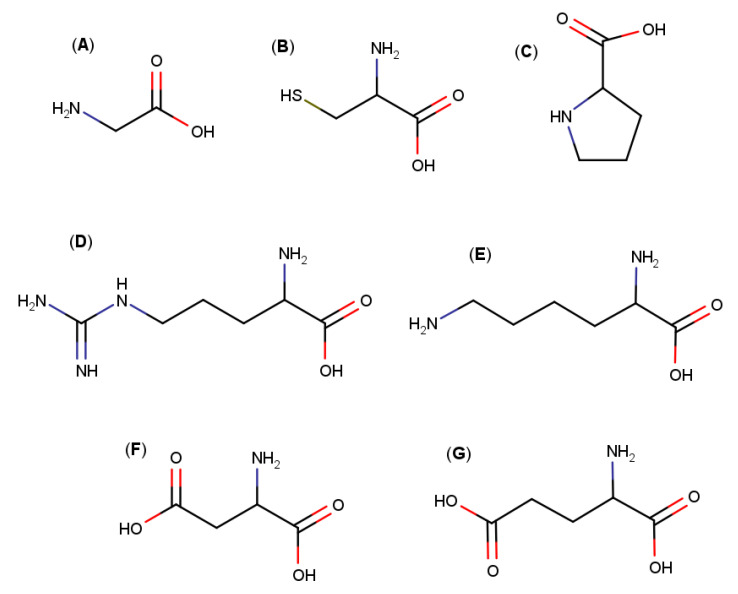
Molecular structures of: (**A**) glycine, (**B**) cysteine, (**C**) proline, (**D**) arginine, (**E**) lysine, (**F**) aspartic acid and (**G**) glutamic acid.

**Figure 3 pharmaceutics-13-01099-f003:**
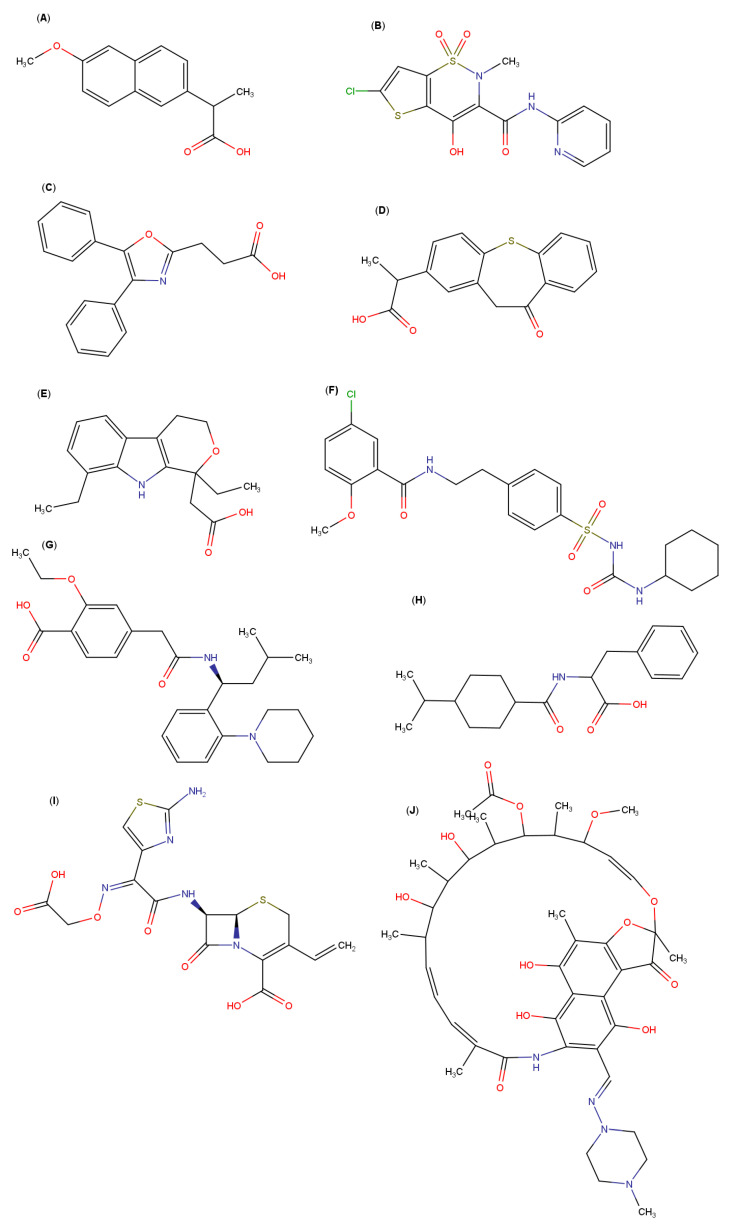
Molecular structures of: (**A**) naproxen, (**B**) lornoxicam, (**C**) oxaprozin, (**D**) zaltoprofen, (**E**) etodolac, (**F**) glyburide, (**G**) repaglinide, (**H**) nateglinide, (**I**) cefixime and (**J**) rifampicin.

**Figure 4 pharmaceutics-13-01099-f004:**
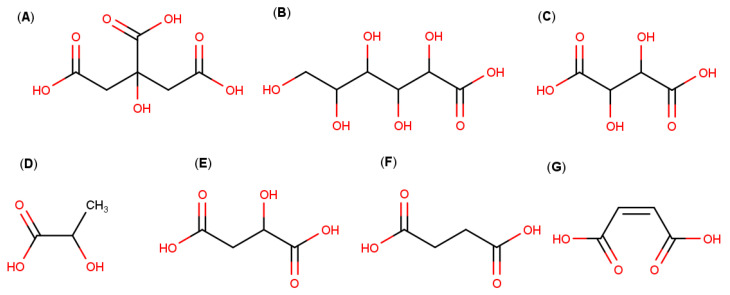
Molecular structures of: (**A**) citric acid, (**B**) gluconic acid, (**C**) tartaric acid, (**D**) lactic acid, (**E**) malic acid, (**F**) succinic acid and (**G**) maleic acid.

**Figure 5 pharmaceutics-13-01099-f005:**
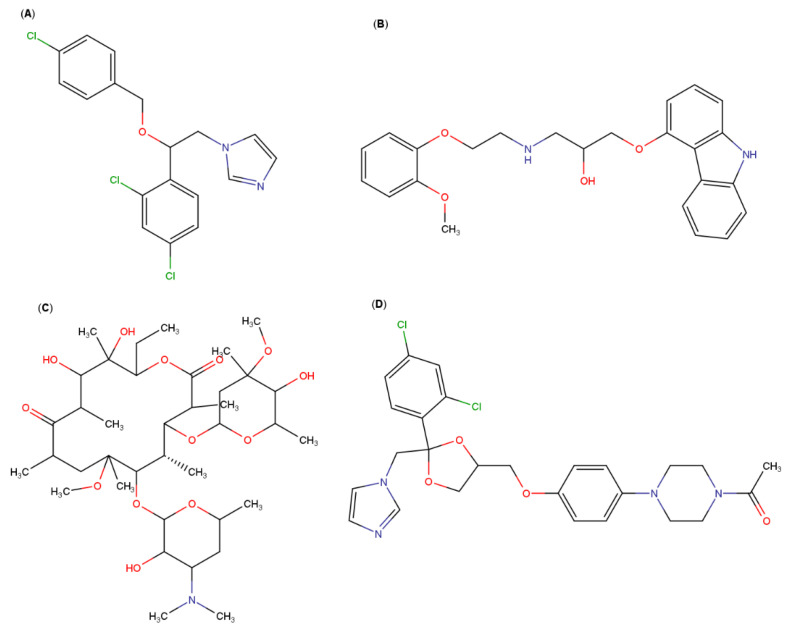
Molecular structures of: (**A**) econazole, (**B**) carvedilol, (**C**) clarithromycin and (**D**) ketoconazole.

**Figure 6 pharmaceutics-13-01099-f006:**
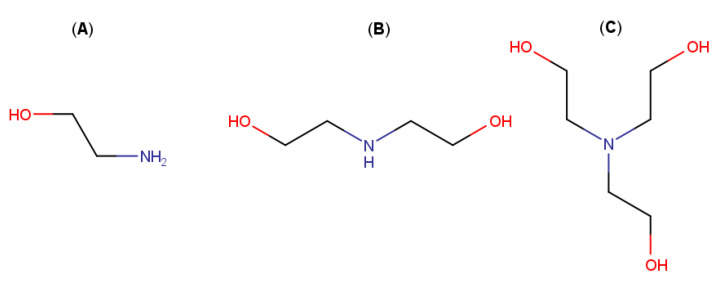
Molecular structures of: (**A**) monoethanolamine, (**B**) diethanolamine and (**C**) triethanolamine.

**Figure 7 pharmaceutics-13-01099-f007:**
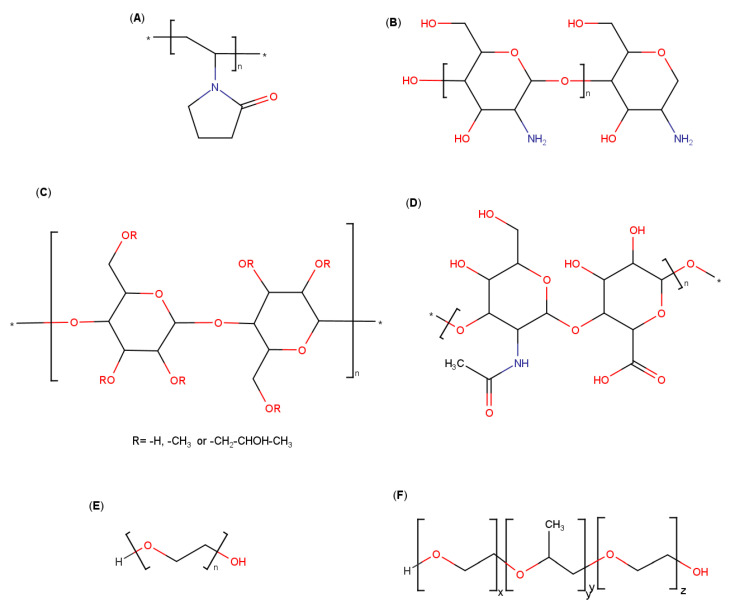
Molecular structures of (**A**) PVP-K30, (**B**) chitosan, (**C**) hydroxypropylmethylcellulose, (**D**) hyaluronic acid, (**E**) polyethylene glycol and (**F**) poloxamer.

**Table 1 pharmaceutics-13-01099-t001:** Summary of some reports on ternary complexes of CDs with arginine, including the improvements in pharmaceutical properties.

Ternary System	Molar Ratio	Preparation Method	Solubility	Dissolution	Other Properties	Reference
naproxen-HPβCD-arginine	1:1:1	co-grinding coevaporation	synergistic action between HPβCD and arginine with 13-fold solubility increment	superior performance of coevaporate complex with 15 times increase in DE		[12,13]
lornoxicam-βCD-arginine	1:2:1	freeze-drying	freeze-dried complex showed the higher solubility saturation in different buffer media	freeze-dried complex exhibits >95% dissolution after 20 min		[14]
oxaprozin-MβCD-arginine	1:1:1	co-grinding coevaporation kneading	4.5-fold solubility increment compared with the binary complex at pH 6.8	co-ground complex showed better performance (DE60 = 83) compared with the oxaprozin:MβCD complex (DE60 = 6.8)		[15]
zaltoprofen-βCD-arginine zaltoprofen-HPβCD-arginine	1:1:1	spray-drying coevaporation	spray-dried ternary complex with HPβCD showed the higher solubility saturation in water and buffer pH 1.2	Both, the spray-dried and co-ground ternary complex with βCD and HPβCD showed 14- and 15-fold increase in dissolution, respectively compared with pure drug		[16]
etodolac-HPβCD-arginine	1:1:1	spray-drying coevaporation	Spray-dried and coevaparate ternary complex showed an increment of 163- and 100-fold, respectively, in water with respect to saturation solubility of pure drug	Spray-dried and coevaparate ternary complex exhibits an increase in percent drug release of 19- and 20-fold, respectively, with respect to pure etodolac.		[17]
glyburide-HPβCD-arginine	1:1:0.5	co-grinding	ternary complex showed higher solubility in both aqueous media and buffer pH 7.5	ternary complex exhibits significant improvement in the dissolution profile compared with the pure drug in unbuffered aqueous media		[20]
repaglinide-HPβCD-arginine	1:1:1	coevaporation kneading	coevaporate complex showed maximum increase in solubility of drug (475-fold)	ternary complexes showed enhanced dissolution rate		[21]
nateglinide-HPβCD-arginine	1:1:1	kneading coevaporation spray-drying	ternary complexes showed higher solubility in both aqueous media and buffer pH 1.2	ternary complexes showed better performance (DE5 = 22) compared with the drug alone (DE5 = 6.6)		[22]
cefixime-βCD-arginine cefixime-HPβCD-arginine		spray-drying	ternary complex showed better drug solubility	ternary complex with HPβCD exhibit better performance (DE2 = 25) compared with the drug alone (DE2 = 0.75)		[23]
rifampicin-βCD- arginine	1:1:1	freeze-drying	ternary complex showed an increase in the drug solubility	ternary complex showed an increase in the dissolved percentage of the drug (53% versus 25% for drug alone)	higher antibiofilm activity with respect to pure rifampicin	[24]

**Table 2 pharmaceutics-13-01099-t002:** A summary of some reports on ternary complexes of cyclodextrins with citric acid (CA), including the pharmaceutical improvements.

Ternary System	Molar Ratio	Preparation Method	Solubility	Dissolution	Other Properties	Reference
Econazole nitrate-SBEβCD-CA	1:1:1	co-grinding	synergistic action between SBEβCD and CA with 21.2 fold solubility increment	superior performance of co-ground system concentration increased to 479.07 ± 11.3 μg/mL after 45 min	higher antimycotic activity with respect to that of pure Econazole	[40,44]
Econazole-αCD-CA	1:1:1		highest solubilizing power with 2480 fold relative solubility increment			[52]
Econazole-γCD-CA	1:1:1		1540 fold relative solubility increment			
Econazole-βCD-CA	1:1:1		2460 fold relative solubility increment			
Econazole-HPβCD-CA	1:1:1		2440 fold relative solubility increment			
Carvedilol-βCD-CA	1:2:2	kneading spray-drying	spray-dried complex exhibits the higher solubility saturation in different buffer media	spray-dried complex exhibits significant improvement in the dissolution profile compared with the pure carvedilol	spray-dried complex is stable in exposition to 40 °C at 75% relative humidity	[45]
Clarithromycin-βCD-CA	1:1:1	freeze-drying		freeze-dried complex exhibits rapid and enhanced dissolution rate in basic media	freeze-dried complex exhibits slightly improved absorption in beagle dogs	[46]
Clarithromycin-βCD-CA		coevaporation freeze-drying	104-fold solubility increment			[54]
Ketoconazole-βCD-CA	1:2:1	spray-drying		spray-dried complex reach a dissolution percentage close to 100%		[55]

## Data Availability

Not applicable.

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
