# Peer review of "Cyclodextrin Multicomponent Complexes: Pharmaceutical Applications"

_pharmaceutics, 2021, doi:10.3390/pharmaceutics13071099_

Round 1

Reviewer 1 Report

The review on multicomponent complexes of cyclodextrins is of high interest. The literature is huge, so a selection and evaluation of the publications makes it easier for the readers to learn the potential of the ternary components. The manuscript is well written and is suggested for publication after some minor corrections.

After the several case studies in conclusions I would expect some advices to the readers which ternary component has to be used for which drugs or which problems to solve.

The 109 references give a good picture on the literature. Still I miss one: it is of high importance that Selva et al could detect gaseous protonated 1:1:1 β-CD-drug-DEA adducts by ionspray ionization and tandem mass spectrometry in 1996 which was the first direct proof of the existence of a ternary complex (Journal of Mass Spectrometry 31, 1364-1370).

Some practical advices on how the ternary complexes are prepared would be also appreciated. As far as I remember in case of polymer as third component a heat treatment is necessary.

Further minor points:

Line 169, 172, 174, 179, 201 and a lot more: gamma (g) not delta (d)

Line 173-174: check the solubility values, they seem to be mixed

Line 175: complexes prepared by precipitation

Line 176: Add reference [26]

Line 188: drugs, such as piroxicam and ketoprofen, (it is better to specify)

Line 272: due to the enhanced

Line 279: 1223 times (I would not use decimals)

Line 316: optimized

Line 319: miconazole (n + e)

Line 322: inclusion

Line 419: slight

Line 433: hydroxypropyl derivative (not hydroxyl)

Line 533: give the meaning of TA

Line 642: obtained..obtaining.. (rephrase)

Line 670: delete This section

Line 672: constant constant

Line 705: check the reference (ref. 76 is on olanzapine not efavirenz)

Line 852: slight

Ref. 34: give volume and page numbers (1176, 470-477)

Ref 37: give article number: 128615

Author Response

- Reviewer #1, Comment #1: After the several case studies in conclusions I would expect some advice to the readers which ternary component has to be used for which drugs or which problems to solve.

Author response: In the revised manuscript the conclusions were revised as suggested by the reviewer and modified as mentioned below:

CDs are excipients widely used by the pharmaceutical industry, which are incorporated in many pharmaceutical formulations marketed in several regions of the world. In recent decades, the development of multicomponent systems has increased the CD potential not only to enhance drug solubility, dissolution rate, and bioavailability but also to allow the modulation of other drug properties such as the stability and biological activity. The CD multicomponent complexation with organic acids or bases, amino acids and polymers prove to be very useful in improving and controlling properties of ionisable, weakly ionisable and non-ionisable drugs. In general, the most suitable auxiliary agent, as well as the appropriate CD, should be selected considering the particular drug physicochemical properties and the characteristics to optimize. For example, in the case of acidic drugs, the selection of the basic auxiliary agent TEA has been demonstrated to significantly enhance both the solubility and the permeability of the drug. In the case of the amino acids as auxiliary agents, arginine is the most used compound since in general it shows a synergistic effect with the CD to enhance the drug solubility. In summary, the analysis of the available literature shows the interest in CD multicomponent complexes as approaches to improve the therapeutic efficacy of drugs.

- Reviewer #1, Comment #2: The 109 references give a good picture on the literature. Still I miss one: it is of high importance that Selva et al could detect gaseous protonated 1:1:1 β-CD-drug-DEA adducts by ionspray ionization and tandem mass spectrometry in 1996 which was the first direct proof of the existence of a ternary complex (Journal of Mass Spectrometry 31, 1364-1370).

Author response: We consider that this interesting work is very important, however we have not included it in this review since it does not refer to modifications about the properties of the drugs included (glibenclamide and furosemide) such as solubility, dissolution rate or stability.

- Reviewer #1, Comment #3: Some practical advice on how the ternary complexes are prepared would be also appreciated. As far as I remember, in case of polymer as the third component a heat treatment is necessary.

Author response: To attend to this suggestion the following paragraph was added to section Introduction:

Regarding the methodological aspects, the techniques commonly used to obtain binary complexes can be used in the preparation of ternary ones in solid state. Methods such as blending, co-grinding, kneading, coevaporation/coprecipitation, freeze-drying, spray-drying and supercritical fluid technology have been used [7]. The method used for the multicomponent complex preparation has a significant impacts on the final product. Depending on the method selected, different parameters should be optimized, such as degree and mixing time, temperature and heating time, among others.

- Reviewer #1, Comment #4: Minor points

Line 169, 172, 174, 179, 201 and a lot more: gamma (γ) not delta (δ)

Line 173-174: check the solubility values, they seem to be mixed

Line 175: complexes prepared by precipitation

Line 176: Add reference [26]

Line 188: drugs, such as piroxicam and ketoprofen, (it is better to specify)

Line 272: due to the enhanced

Line 279: 1223 times (I would not use decimals)

Line 316: optimized

Line 319: miconazole (n + e)

Line 322: inclusion

Line 419: slight

Line 433: hydroxypropyl derivative (not hydroxyl)

Line 533: give the meaning of TA

Author response: The meaning of TA appears on line 414 of the original manuscript.

Line 642: obtained..obtaining.. (rephrase)

Line 670: delete This section

Line 672: constant constant

Line 705: check the reference (ref. 76 is on olanzapine not efavirenz) This reference was checked and changed by the correct cite.

Line 852: slight

Ref. 34: give volume and page numbers (1176, 470-477)

Ref 37: give article number: 128615

Author response: We performed all the suggested corrections in the manuscript.

Author Response

- Reviewer #2, Comment #1:  The authors should add a paragraph highlighting the novelty of this work and its contribution in the field. Different classes of auxiliary substances that can be used in conjunction with cyclodextrin:drug complexes are presented and several studies related to the use of these substances as auxiliary agents are summarized.

Author response: the following paragraph of the Introduction was modified to highlight our contribution in the field:

“Based on the suitability of amino acids, organic acids and bases, and water-soluble polymers as auxiliary agents with some advantages over other excipients, this article reviews and tries to cover the most relevant CD multicomponent complexes obtained with this auxiliary compounds as well as their use in the pharmaceutical field. Their application in drug delivery is illustrated and discussed in depth by specific examples reported in the literature to support the benefits of their application in the development of drug formulations with improved properties.”

- Reviewer #2, Comment #2:  1. Figure 1 is not properly cited to the original source of the image.

Author response: The following information was included in the legend of figure 1:

“The image was prepared using Pymol v.2.1 and in-house developed Python scripts.”.

- Reviewer #2, Comment #3:  Figure 1 is also not referenced anywhere in text.

Author response:  The figure 1 is referenced in the introduction section of the manuscript:

“The most common natural CDs are αCD, βCD and γCD (Figure 1), consisting of 6, 7 and 8 d-glucose units, and are the only ones used for pharmaceutical applications”.

- Reviewer #2, Comment #4:  Heading 5 (Water-soluble polymers as auxiliary agents) may be divided into sub-headings for each polymer, as it currently lacks proper continuity between paragraphs.

Author response: Sub-headings for each polymer were added.

- Reviewer #2, Comment #5:  Molecular structures of polymers in Figure 5 are incorrect (Hydroxypropylmethylcellulose, Hyaluronic acid, Polyethylene glycol). Structures are also not drawn in a consistent style.

Author response: Structures now were drawn with a consistent style.

- Reviewer #2, Comment #6:  Figures and tables could be included that show relevant results of certain studies, or show comparisons between different types of auxiliary agents.

Author response: Tables 1 and 2 showing relevant information of some auxiliary agents were included in the revised version of the manuscript.

Reviewer 3 Report

The minireview of Aiassa et al. on the multicomponent cyclodextrin complexes could become useful for pharmacists, formulation developers, and managers when the manuscript reaches an acceptable quality. The current form contains many intricated texts, wordy sentences, typos, and sometimes the US and British spelling is mixed, and please decide which spelling is the more convenient and stick to that in the manuscript. A major review can correct the actual weaknesses. Before the resubmission, the help of a native English lector is more than advisable.

Below, without completeness, some major points are mentioned.
- Much incorrect punctuation, missing super- (M-1), and subscripts (Cmax/Tmax) are in the text! Two decimal digits of the hundred-range - or thousand (!), line 279 - stability constants are funny.

- The authors have created schemes for many common molecules but missed them for the mentioned guest molecules. Perhaps the structures of the complexed molecules are less known than, e.g., glycine or ethanolamines. Please, provide the structures of the guest molecules mentioned in the manuscript.

- The Conclusion is very general. Please insert some information and relevant conclusions into this section.

- The Abstract is nothing. The first two sentences are excerpts from the Introduction only. Please, create an informative abstract.

- Please give a reason for the repetition of "cyclodextrin" and "multicomponent systems" from the title in the Keyword section. As far as the referee knows, the Keyword section is a separate field.

- In line 34, the "hydrophobic interior" of the CD (perhaps the authors mean the cavity) is not correct. The interior space is only less hydrophilic than the hydroxyl rims or the aqueous bulk solution. The whole manuscript, by the verification of cyclodextrins, rebutted this statement. Additionally, the native CDs contain water molecules inside the cavity, even in the crystalline state, and water re-adsorbs fast after complete drying. Water is a hydrophilic molecule.

- The constituting units are alpha-D-glucopyranosides and not glucopyranoses (line 37). The glycosidic carbohydrate derivatives end with "ide" and not "ose" because the latter indicates the presence of a reducing end.

- Among the native CDs, only the beta version has low solubility (~1.8-2%, depending on the dried or shelf-dry state). Line 38.

Line 43, the correct name of hydroxypropyl is (2-hydroxy)propyl.  While it is true that, especially with the many pharmaceutical publications, the incorrect version is flooding the technical pages.

- Line 57, the sugammadex (ORG-25969) was synthesized and introduced by Organon and not MSD. Because this derivative is one of the pioneers in the anticurare complexation, the originator deserves a better mention than a multinational supplier. Additionally, a reference is missing.

- Figure 1 is interesting but meaningless without mentioning the meaning of colors. Are the colors representing charges, lipophilicity, atoms, or what? And what do the surfaces represent? Are they the van der Waals or solvent (water) accessible surfaces? In addition, there is no reference to the software used to create the figures.

- Line 94 contains a typo.

- Lines 311-313 are useless.

- Line 319 seems to have a typo, or the authors mixed a commercial name with the ingredient.

- Line 376, "un-ionized" is unintelligible.

- Line 457 is in the section of basic auxiliaries, but the reference principally deals with ascorbic acid and its TEA salt. Please, reconsider its location.

- What is the difference between the randomly methylated beta-cyclodextrin (RMßCD) and the otherwise identical methyl beta-cyclodextrin (MeßCD)?

- Figure 4 is useless.

- Line 757 contains a typo (an arrow in β-(1-4)D-).

- Line 778 and 779, incorrect spelling.

- What is RPE? Please, scan the manuscript for unresolved abbreviations.

Author Response

- Reviewer #3, Comment #1:  A major review can correct the actual weaknesses. Before the resubmission, the help of a native English lector is more than advisable.

Author response: the manuscript was revised before the resubmission.

- Reviewer #3, Comment #2:  Much incorrect punctuation, missing super- (M-1), and subscripts (Cmax/Tmax) are in the text! Two decimal digits of the hundred-range - or thousand (!), line 279 - stability constants are funny.

Author response: The punctuation, super- and subscripts were revised and changed.

- Reviewer #3, Comment #3:  The authors have created schemes for many common molecules but missed them for the mentioned guest molecules. Perhaps the structures of the complexed molecules are less known than, e.g., glycine or ethanolamines. Please, provide the structures of the guest molecules mentioned in the manuscript.

Author response: The structures of some guest drugs, that were used in the preparation of complexes, were included in tables 1 and 2.

- Reviewer #3, Comment #4:  The Conclusion is very general. Please insert some information and relevant conclusions into this section.

Author response: In the revised manuscript the conclusions were revised as suggested by the reviewer and modified as mentioned below:

CDs are excipients widely used by the pharmaceutical industry, which are incorporated in many pharmaceutical formulations marketed in several regions of the world. In recent decades, the development of multicomponent systems has increased the CD potential not only to enhance drug solubility, dissolution rate, and bioavailability but also to allow the modulation of other drug properties such as the stability and biological activity. The CD multicomponent complexation with organic acids or bases, amino acids and polymers prove to be very useful in improving and controlling properties of ionisable, weakly ionisable and non-ionisable drugs. In general, the most suitable auxiliary agent, as well as the appropriate CD, should be selected considering the particular drug physicochemical properties and the characteristics to optimize. For example, in the case of acidic drugs, the selection of the basic auxiliary agent TEA has been demonstrated to significantly enhance both the solubility and the permeability of the drug. In the case of the amino acids as auxiliary agents, arginine is the most used compound since in general it shows a synergistic effect with the CD to enhance the drug solubility. In summary, the analysis of the available literature shows the interest in CD multicomponent complexes as approaches to improve the therapeutic efficacy of drugs.

- Reviewer #3, Comment #5:  The Abstract is nothing. The first two sentences are excerpts from the Introduction only. Please, create an informative abstract.

Author response: In the revised manuscript the abstract was revised as suggested by the reviewer and modified as mentioned below:

“Cyclodextrins (CDs) are naturally available water-soluble cyclic oligosaccharides widely used as carriers in the pharmaceutical industry for their ability to modulate several properties of drugs through the formation of drug:CD complexes. The addition of an auxiliary substance to form multicomponent complexes is an adequate strategy to enhance complexation efficiency and to facilitate the therapeutic application of different drugs. This review discusses the multicomponent complexation using amino acids, organic acids and bases, and water-soluble polymers as auxiliary excipients. Special attention is given to improved properties such as solubility, dissolution, permeation, stability and bioavailability of several relevant drugs. In addition, the use of CD multicomponent complexes to enhance therapeutic drug effects is summarized.”

- Reviewer #3, Comment #6:  Please give a reason for the repetition of "cyclodextrin" and "multicomponent systems" from the title in the Keyword section. As far as the referee knows, the Keyword section is a separate field.

Author response: we replace the words "cyclodextrin" and "multicomponent systems" with the words auxiliary agents and complexation efficiency.

- Reviewer #3, Comment #7: In line 34, the "hydrophobic interior" of the CD (perhaps the authors mean the cavity) is not correct. The interior space is only less hydrophilic than the hydroxyl rims or the aqueous bulk solution. The whole manuscript, by the verification of cyclodextrins, rebutted this statement. Additionally, the native CDs contain water molecules inside the cavity, even in the crystalline state, and water re-adsorbs fast after complete drying. Water is a hydrophilic molecule.

Author response: the sentence of the original manuscript was replaced by the following one: “Their structure resembles a truncated cone with a somewhat lipophilic central cavity and external hydrophilic surface.”

- Reviewer #3, Comment #8:  The constituting units are alpha-D-glucopyranosides and not glucopyranoses (line 37). The glycosidic carbohydrate derivatives end with "ide" and not "ose" because the latter indicates the presence of a reducing end.

Author response: the sentence of the original manuscript was replaced by the following one: “The most common natural CDs are αCD, βCD and γCD (Figure 1), consisting of 6, 7 and 8 d-glucose units, and are the only ones used for pharmaceutical applications.”

- Reviewer #3, Comment #9: Among the native CDs, only the beta version has low solubility (~1.8-2%, depending on the dried or shelf-dry state). Line 38.

Author response: We modify the sentence.

- Reviewer #3, Comment #10:  Line 43, the correct name of hydroxypropyl is (2-hydroxy)propyl.  While it is true that, especially with the many pharmaceutical publications, the incorrect version is flooding the technical pages.

Author response: We performed the suggested correction.

- Reviewer #3, Comment #11:  Line 57, the sugammadex (ORG-25969) was synthesized and introduced by Organon and not MSD. Because this derivative is one of the pioneers in the anticurare complexation, the originator deserves a better mention than a multinational supplier. Additionally, a reference is missing.

Author response: We performed the suggested corrections.

“It should be noted that the modified γ-CD sugammadex developed by the pharmaceutical company Organon and currently marketed by Merck Sharp & Dohme (Bridion®) is used as a drug substance indicated to reverse the neuromuscular blockade induced by rocuronium and vecuronium in adults undergoing surgery [4].”  

- Reviewer #3, Comment #12:  Figure 1 is interesting but meaningless without mentioning the meaning of colors. Are the colors representing charges, lipophilicity, atoms, or what? And what do the surfaces represent? Are they the van der Waals or solvent (water) accessible surfaces? In addition, there is no reference to the software used to create the figures.

Author response:  To attend to this suggestion the following information was added in Legend of Figure 1:

“The surfaces shown correspond to the solvent-accessible surface area (SASA) correspond-ing to a water molecule, with colours by atom-type indicating the region of the molecule that is accessible (red: oxygen and grey: carbon atoms). SASA was calculated using cy-clodextrin structures deposited in the Cambridge Crystallographic Data Centre under 1100537, 1107195 and 1126611 deposition numbers for αCD, βCD and γCD, respectively. The image was prepared using Pymol v.2.1 and in-house developed Python scripts.”

- Reviewer #3, Comment #13:  Line 94 contains a typo.

Author response: We performed the suggested correction.

- Reviewer #3, Comment #14:  Lines 311-313 are useless.

Author response: We consider that it is important to mention the interaction modes between the molecules because it contributes to the complex formation.

- Reviewer #3, Comment #15:  Line 319 seems to have a typo, or the authors mixed a commercial name with the ingredient.

Author response: In the revised manuscript “micomazol” was changed by “miconazole”.

- Reviewer #3, Comment #16:  Line 376, "un-ionized" is unintelligible.

Author response: We performed the suggested correction.

- Reviewer #3, Comment #17:  Line 457 is in the section of basic auxiliaries, but the reference principally deals with ascorbic acid and its TEA salt. Please, reconsider its location.

Author response: In this case we considered ascorbic acid as the guest molecule and TEA as a basic auxiliary agent.

- Reviewer #3, Comment #18:  What is the difference between the randomly methylated beta-cyclodextrin (RMßCD) and the otherwise identical methyl beta-cyclodextrin (MeßCD)?

Author response: In the revised manuscript “RMβCD” was changed by MeβCD.

- Reviewer #3, Comment #19:  Figure 4 is useless.

Author response: we consider that figure 4 may be useful for some readers.

- Reviewer #3, Comment #20:  Line 757 contains a typo (an arrow in β-(1-4)D-).

Author response: this arrow is not in the original manuscript, perhaps it was misconfigured when going to the journal template.

- Reviewer #3, Comment #21:  Line 778 and 779, incorrect spelling.

Author response: The spelling was corrected.

- Reviewer #3, Comment #22:  What is RPE? Please, scan the manuscript for unresolved abbreviations.

Author response: RPE is retinal pigment epithelial (RPE) cell line. This was resolved in the manuscript.

Round 2

Reviewer 2 Report

The authors have addressed the previous comments adequately. However, chemical structures provided in table 1 and 2 are very blurred and must be improved before accepting the manuscript in this journal.

Author Response

Author response: the chemical structures were improved.

Reviewer 3 Report

The revision has significantly improved the manuscript of Aissa et al., and only a few minor corrections seem necessary.

Line 37 still contains "glucopyranose" and not "pyranoside".

Line 42, "d-glucose" should be "D-glucose".

Line 64, although the mention of the current supplier of sugammadex is still hardly understood - except if it is the sponsor of this manuscript.
As for reference 4, citation of the original article would be better than referencing a general, otherwise excellent book (A. Bom, M. Bradley, K. Cameron, J.K. Clark, J. Van Egmond, H. Feilden, E.J. MacLean, A.W. Muir, R. Palin, D.C. Rees, M.-Q. Zhang, A novel concept of reversing neuromuscular block: chemical encapsulation of rocuronium bromide by a cyclodextrin-based synthetic host. Angew Chem Int Ed Engl., 2002, 41(2), 266-70. doi: 10.1002/1521-3773(20020118)41:2<265::aid-anie265>3.0.co;2-q). The referee assumes that the authors of the current manuscript would also be less satisfied if they were omitted from a review article as authors of a truly ground-breaking communication. But this is up to the authors, and if they see that the supplier's name is important, the reviewer should accept it.
It is necessary to mention that the reviewer's comment is for fairness, and the reviewer has/had no connection or any common interest with the inventors of sugammadex.

Author Response

Author response: We performed the suggested correction in the manuscript.

Author response: We performed all suggested corrections in the manuscript. We further wish to point out that the current supplier of sugammadex is not a sponsor of this manuscript nor have we had any relationship with them or with said product.
